# Educational Needs of Nurses for Respiratory Communicable Infectious Disease Care: A Cross-Sectional Descriptive Study

**DOI:** 10.3390/healthcare9081043

**Published:** 2021-08-13

**Authors:** Jeong-Won Han, Ji-Soon Kang, Jun-Hee Park

**Affiliations:** 1College of Nursing Science, Kyung Hee University, 26 Kyunghee-daero, Dongdaemun-gu, Seoul 02447, Korea; hjw0721@naver.com; 2Department of Nursing, Hansei University, 30 Hansei-ro, Gunpo-si 43742, Korea; lemueljs@gmail.com; 3Department of Nursing, Dongnam Health University, 50 Cheoncheon-ro 74 gil, Jangan-gu, Suwon-si 16328, Korea

**Keywords:** education, need, nurse

## Abstract

Clinical nurses have training needs related to the strategies for infection spread prevention. This study aimed to investigate the levels of importance of and performance in the various areas of care for communicable respiratory infections among clinical nurses and to determine the priority of educational needs. Hospitals in which a baseline survey could be conducted were considered, and nurses working in six hospitals that were designated as COVID-19 care centers in Korea were enrolled. The training needs for the care of patients with communicable respiratory infections were analyzed using Borich’s needs equation, and the locus for focus model. Among participants with prior COVID-19 patient care experience, according to Borich’s equation, the need score was the highest for “initial response to communicable respiratory infection”, followed by “management of aerosol-generating procedures in patients with communicable respiratory infection” and “reporting of patients with communicable respiratory infection and death of patient”. An item rated highly in both Borich’s equation and the locus for focus model among the participants with prior experience was “initial response to communicable respiratory infection”. Among participants without prior relevant experience, according to Borich’s equation, the need score was highest for “management of aerosol-generating procedures in patients with a communicable respiratory infection”, followed by “initial response to communicable respiratory infection” and “reporting of patients with communicable respiratory infection and death of patient”. None of the items were rated high in both Borich’s equation and the locus for focus model among participants without relevant prior experience.

## 1. Introduction

COVID-19 is a novel communicable infection that poses a threat to public health [1,2]. The COVID-19 pandemic has created many challenges for health systems across the world [3]. One of the less discussed topics relates to the rapid need for training critical or noncritical care nursing staff to take care of COVID-19 patients [4,5]. Healthcare providers are at a high risk of acquiring infection during a novel communicable infection outbreak through the provision of patient care and treatment [6]. Nurses, in particular, are in direct contact with patients and are exposed to various samples and contaminated medical equipment as an occupational hazard. Additionally, they tend to work in environments with a high risk of infection due to personal protective equipment (PPE) shortages and a lack of accurate protocols [7]. As a result, nurses experience fear pertaining to their health and that of their families, anxiety, work overload, and ethical dilemmas in the course of the provision of care to patients with an infectious disease [8]. While 15 nurses who had been providing care to infected patients during the MERS-CoV epidemic acquired the infection from these patients, 25 physicians and 190 nurses have acquired COVID-19 from patients in the current pandemic, leading to escalated anxiety and fear levels [2]. According to an analysis of the response to MERS-CoV, a lack of knowledge and information on MERS-CoV, contamination during PPE removal, and a lack of PPE education and training were identified as some causes of infection among healthcare providers, highlighting the urgent need for the provision of relevant education [9]. A previous study on MERS-CoV [10] reported that Korean nurses had a high level of knowledge on transmission prevention and disease characteristics but a low level of knowledge on diagnosis, treatment, and PPE use, with the greatest educational needs pertaining to the transmission routes and prevention of the spread of MERS-CoV and relevant quarantine guidelines. In improving the level of compliance among nurses in the setting of novel communicable diseases, the provision of appropriate education, through systematic and continuous programs that enable repeated learning through indirect experiences, is crucial [11]. Further, for the development of such education programs and to boost their effectiveness, it is important to analyze the characteristics and needs of the learners. Educational needs analysis involves the examination of the gap between the importance of a particular educational area and a learner’s current performance in that area [12]. In educational needs, “importance” refers to the desired level of competency, that is, this is “what should be” for a competency that is essential and desired. “Current performance” refers to “what is”, as determined by learners in their current states, and denotes the level of self-efficacy in translating their competency into practice [13]. An analysis of educational needs identifies the gap between these two levels. The training needs assessment equation developed by Borich [14] requires learners to self-evaluate the importance and current performance for each competency and weigh the gap between the two. The results are listed in order, based on which the priority of the competency areas can be established [15]. Thus, this study aimed to investigate the levels of importance and performance in the various areas of care for communicable respiratory infection among clinical nurses and determine the priority in training needs using the Borich training needs equation (Figure 1) and the locus for focus model (Figure 2). Furthermore, this study was conducted to provide basic data for the establishment of comprehensive education and training strategies for nurses who are newly introduced to the care of COVID-19 patients.

## 2. Materials and Methods

### 2.1. Study Location and Participant

This study was cross-sectional descriptive research. Hospitals in which a baseline survey could be conducted were considered for nurses working in six hospitals that were designated as COVID-19 care centers. The sample size was determined using G-power 3.1.10 software. Based on a study that examined nurses’ training needs for MERS management [10], the minimum sample size for a single group was determined to be 144, at a significance level (α) of 0.05, effect size of 0.3, and power (1-β) of 0.90 for one-way analysis of variance. As this study compared the difference in the training needs between two groups divided based on their experience in caring for patients with communicable respiratory infections, at least a total of 288 participants was required to allow for the assignment of 144 participants to each group. A total of 219 participants experienced in COVID-19 patient care and 153 without the relevant experience were enrolled; thus, the minimum sample size was met. A total of 219 participants (58.9%) had prior experience with COVID-19 patient care, while 153 (41.1%) lacked the relevant experience. The group with prior experience showed female predominance (female sex, *n* = 206, 94.1% versus male sex, *n* = 13, 5.9%) and the mean age was 35.14 ± 8.98 years. The marital status in this group was either “married” (*n* = 111, 50.7%) or “single and other” (*n* = 108, 49.3%). A majority of these participants had a Bachelor’s degree (*n* = 157, 71.7%) and the mean total clinical career duration was 125.83 ± 97.85 months. The mean career duration in the current unit was 46.26 ± 56.62 months. A majority of the participants (*n* = 159, 72.6%) worked in general wards, including respiratory and infectious disease wards. The mean number of infection management training sessions attended was 3.33 ± 2.89, with each session lasting longer than 60 min in 108 participants (49.3%). The most commonly used learning method for infection management was clinical training (*n* = 131, 59.8%), while the most effective, as perceived by the participants, was a combination of lectures and training (*n* = 131, 59.8%). The group without prior COVID-19 patient care experience also showed female predominance (female, *n* = 134, 87.6% versus male, *n* = 19, 12.4%) and the mean age was 32.54 ± 8.19 years. The marital status in this group tended to be either “married” (*n* = 50, 32.7%) or “single and other” (*n* = 103, 67.3%). A majority of the participants without relevant experience had a Bachelor’s degree (*n* = 112, 73.2%) and their mean total clinical career duration was 98.85 ± 90.88 months. The mean career duration in the current unit was 42.21 ± 48.98 months, and a majority of the participants (*n* = 99, 64.7%) worked in a general ward. The mean number of infection management training sessions was 3.24 ± 2.45, with each session lasting longer than 60 min in 75 participants (49.0%). The major learning method for infection management was clinical training (*n* = 72, 47.1%), while the most effective, as perceived by the participants, was a combination of lectures and training (*n* = 78, 51.0%) (Table 1).

### 2.2. Instruments

Training needs refer to nurses’ perceived level of needs pertaining to knowledge and skills related to patients’ health, disease prevention, and health promotion [16]. In this study, training needs referred to the perceived importance score and self-efficacy score in relation to the provision of care for communicable respiratory infection using a tool we developed based on MERS [17] and COVID-19 [2] response guidelines, which was validated by a panel of experts. The specific competency areas of training needs were: disease characteristics, diagnosis and testing, treatment method, transmission route and spread prevention, use of PPE, quarantine guidelines, repeated use of equipment for inpatients, requirements to end patient isolation, ward environmental management, disposal of medical waste, and nurses’ personal hygiene. The perceived importance and self-efficacy for each competency were rated on a 5-point Likert scale.

### 2.3. Data Collection

Data were collected from six hospitals from 2 September 2020 to 31 October 2020. Prior to data collection, this study was approved by the institutional review board (KHSIRB-20-330(EA)). Further, data were collected only from hospitals that granted permission for the performance of a baseline survey amid the COVID-19 pandemic. We obtained informed consent from the participants, who were given an information sheet that provided clarity on the purpose and method of the study, anonymity, research use of data, and freedom to withdraw from the study at any time.

### 2.4. Data Analysis

The participants’ general characteristics were analyzed using frequency, percentage, and mean with standard deviation. The training needs for the care of patients with communicable respiratory infections were analyzed using paired sample *t*-tests, Borich’s needs equation, and the locus for focus model. In the first step of the analysis, the participants were divided into two groups based on their prior experience with COVID-19 patient care. In the second step, Borich’s needs equation was used to determine the priority of the training needs in order to address the shortcoming of the *t*-test that simply compares values. Borich [14] argued that training needs can be identified by analyzing the discrepancy between the current performance of an education program and the desired level of performance. According to Borich’s needs equation, a participant’s performance score is subtracted from the importance score for each competency and the difference is multiplied by the mean importance of the competency and divided by the total number of cases. This generates the needs score, with a higher needs score indicating a higher priority for a particular competency. Finally, to complement and visualize the training needs obtained by Borich’s equation, the locus for focus model was used to determine the coordinate plane. This model plots scores in a two-axis coordinate plane and, thus, is useful for priority visualization. The horizontal axis in the locus for focus model denotes the importance of a competency and the vertical axis represents the discrepancy between importance and current performance efficacy [18]. In the present study, we considered the competencies plotted in quadrant 1 as those with the high training needs, with a high perceived importance and high discrepancy level. In the final step, the competency items present in quadrant 1 (HH) of the locus for focus model and that were determined to have high priority in Borich’s equation were considered as the items with the highest priority.

## 3. Results

### 3.1. Analysis of Training Needs and Priority of Training among Participants with Prior COVID-19 Patient Care Experience

Table 2 shows the results of the training needs assessment among participants with prior COVID-19 patient care experience. The discrepancy between importance and performance was statistically significant for all the competency items. According to Borich’s equation, the need score was the highest for “Initial response to communicable respiratory infection” (3.75), followed by “Management of aerosol-generating procedures in patients with communicable respiratory infection” (3.58), and “Reporting of patients with communicable respiratory infection and death of patient” (3.04). Figure 3 shows the visualization of the priorities using the locus for focus model. In this model, the mean importance of the competency items in the care of communicable respiratory infection patients was 4.66 and the mean discrepancy between importance and performance was 0.60. The items in quadrant 1 (HH) in the locus for focus model were “Characteristics of communicable respiratory infections” and “Initial response to communicable respiratory infection” (Table 2). Lastly, an item rated highly in both Borich’s equation and the locus for focus model among the participants with prior experience was “Initial response to communicable respiratory infection.”

### 3.2. Analysis of Training Needs and Priority of Training among Participants without Prior Experience in COVID-19 Patient Care

Table 3 shows the results of the training needs assessment conducted among the participants without prior relevant experience. The discrepancy between importance and performance was statistically significant for all the competency items. According to Borich’s equation, the need score was the highest for “Management of aerosol-generating procedures in patients with communicable respiratory infection” (5.04), followed by “Initial response to communicable respiratory infection” (4.49), and “Reporting of patients with communicable respiratory infection and death of patient” (4.34). Figure 4 shows the visualization of the priorities using the locus for focus model. In this model, the mean importance of the competency items in the care of communicable respiratory infection patients was 3.77 and the mean discrepancy between importance and performance was 0.86. Only one item was in quadrant 1 (HH) of the locus for focus model: “Testing and management of patients with communicable respiratory infection” (Table 3). Lastly, none of the items were rated high in both Borich’s equation and the locus for focus model among participants without relevant prior experience.

## 4. Discussion

This study aimed to investigate Korean nurses’ perceived importance and confidence in competency area performance in the provision of care to communicable respiratory infection patients with the aim of prioritizing their training needs. First, we found that all our participants perceived the competency areas for the care of communicable respiratory infection patients as important but had relatively low confidence levels in performance. The high perceived importance may be attributable to the global COVID-19 pandemic and the consequent elevated possibility of clinical nurses providing care to COVID-19 patients. The level of confidence in performance was higher among those with prior COVID-19 patient care experience (4.05) than in those without prior relevant experience (3.77), suggesting that the former group performs better in infection management. However, it is important to note that even nurses with prior experience in COVID-19 patient care rated their performance to be lower than their perceived importance for all the competency items. Although several countries are stepping up in their response to COVID-19, healthcare providers are still vulnerable to fear and anxiety due to the threat to their personal safety, inadequate and limited resources, and lack of an accurate protocol [19]. With the prolongation of the COVID-19 pandemic, nurses who have low confidence levels in their performance may only passively respond to situations, calling for healthcare facilities to provide intensive training in communicable respiratory infection settings for nurses.

Second, the participants who had prior COVID-19 patient care experience had the strongest needs for “Initial response to communicable respiratory infection”, “Management of aerosol-generating procedures in patients with communicable respiratory infection”, and “Reporting of patients with communicable respiratory infection and death of patient” in Borich’s training needs assessment. In contrast, those without prior experience had the strongest need for “Management of aerosol-generating procedures in patients with communicable respiratory infection”, “Initial response to communicable respiratory infection”, and “Reporting of patients with communicable respiratory infection and death of patient.” In other words, clinical nurses showed a strong need for training pertaining to the initial response to communicable respiratory infection, management of aerosol-generating procedures, and measures to be taken after patient death. This is similar to the results obtained in a Spanish study on clinical nurses in the emergency department [5], in which nurses experienced early response-related difficulties during the COVID-19 pandemic and a fear of virus transmission, and strongly requested the provision of accurate information on communicable respiratory infections, guidelines, and PPE supply. Furthermore, a study that analyzed articles related to end-of-life care during an epidemic [20] reported that nurses have the challenge of having to engage in infection management in addition to routine end-of-life care. In particular, South Korea declared COVID-19 a first-grade infectious disease amid the escalating number of critically ill patients and related deaths, which imposed an increased burden of administrative procedures on healthcare providers. Therefore, communicable respiratory infection education for clinical nurses should provide accurate information on the protocols for initial response, infection management during aerosol-generating procedures, and processes after a patient’s death. Additionally, healthcare facilities must develop and implement such education programs.

Third, “Characteristics of communicable respiratory infections” and “Initial response to communicable respiratory infection” were in quadrant 1 (HH) in the locus for focus model among participants with prior experience with COVID-19 care, while “Testing and management of patients with communicable respiratory infection” was the only item in quadrant 1 (HH) of the model among those without prior experience. These results are similar to the findings of a previous study conducted among nurses who were in direct contact with COVID-19 patients [21], which found that nurses dealing with communicable infections must strive to provide care while effectively controlling the spread of the infection based on accurate information and knowledge of the disease. In particular, our results showed that nurses with prior COVID-19 patient care experience perceived that having an accurate understanding of the disease and initial response is important, but that their current level of knowledge and performance falls short of the ideal. A majority of people who acquire infections or have symptoms are treated at healthcare facilities and their outcomes differ according to the readiness of healthcare providers, including nurses [22]. Thus, healthcare facilities must prioritize the provision of systematic education to clinical nurses to help them adequately understand the characteristics of communicable respiratory infections and train them to be equipped with the competency required to effectively respond to an infection upon onset.

In contrast, nurses without prior COVID-19 patient care experience had the strongest need pertaining to “Testing and management of patients with communicable respiratory infection”, suggesting that they consider the manner in which newly admitted inpatients or outpatients who are suspected of having a communicable respiratory infection are dealt with as important. However, one notable finding of this study is that none of the training needs overlapped in quadrant 1 (HH) of the locus for focus model and top ranks in Borich’s assessment conducted among nurses without prior experience; hence, the training needs with the highest priority could not be determined. Nurses who have provided care to COVID-19 patients in the past are able to identify the aspects of care that are important and their current level of performance in those areas, as they repeatedly receive infection-related training and provide care to actual patients. On the other hand, nurses without such experience cannot make accurate judgments about the current situation, hindering the identification of the final training needs. However, considering the burgeoning number of confirmed infection cases and critically ill patients, as well as the increasing number of confirmed cases in the community, the role of care provision to communicable respiratory infection patients could extend to the entire pool of healthcare providers. Thus, there is an urgent need for national and organizational efforts to provide information and disseminate skills to help nurses flexibly handle communicable respiratory infections in consideration of their work environments and needs.

Although the results of this study concern one specific area in a country, it can be generalized to other areas and countries worldwide. The current COVID-19 pandemic is a fresh example of how a local threat can become a world issue [23]. Clinical nurse preparedness for caring for confirmed COIVD-19 patients is important for the maintenance and collapse prevention of health systems [24]. It can be confirmed through this study that programs and education tailored to the educational needs of nurses are an important way to reduce the burden on nurses taking care of patients with COVID-19 and prevent the collapse of the national medical system.

## 5. Limitations

Some limitations of this study are that we could not examine the varying educational needs priorities by clinical career among nurses or analyze the varying needs of nursing units. Thus, subsequent studies should address these factors and reflect diverse situations and characteristics for the development of various programs based on the findings. The data of this study is also a limitation of the study due to the self-reporting of subjects.

## 6. Conclusions

In summary, this study showed that Korean nurses have the strongest training needs in the areas related to the characteristics of communicable respiratory infections, initial response, and strategies for infection spread prevention. This study is significant in that it identified the specific educational needs pertaining to the care of communicable respiratory infections for clinical nurses in consideration of the persistent COVID-19 pandemic. Another strength of this study is that the findings can be utilized in the provision of infection control education for nursing students.

## Figures and Tables

**Figure 1 healthcare-09-01043-f001:**
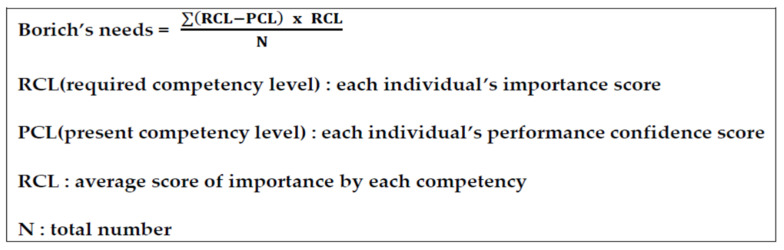
The formula for the priority decision of Borich.

**Figure 2 healthcare-09-01043-f002:**

The locus for focus model.

**Figure 3 healthcare-09-01043-f003:**
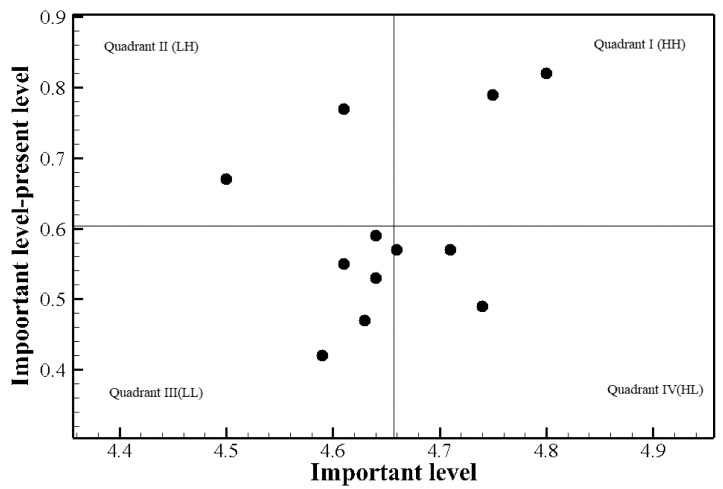
The Locus for Focus model analysis of participants with prior COVID-19 patient care experience.

**Figure 4 healthcare-09-01043-f004:**
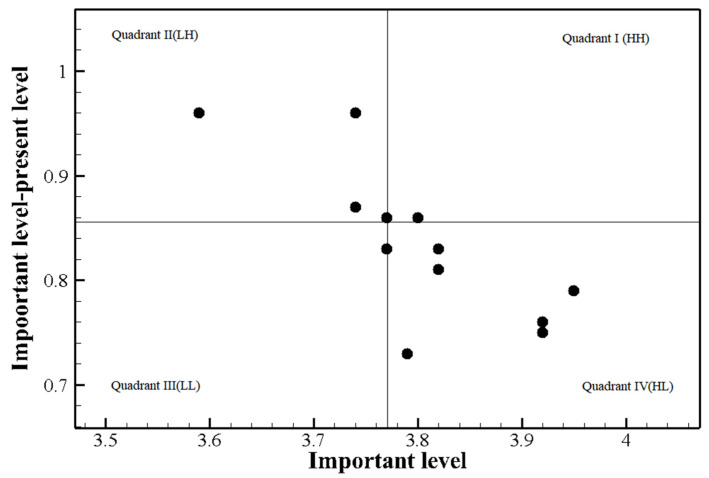
The Locus for Focus model analysis of participants without prior COVID-19 patient care experience.

**Table 1 healthcare-09-01043-t001:** Characteristics of subjects.

Characteristic	Category	Participants with Prior Experience in COVID-19 Patient Care (*n* = 219)	Participants without Prior Experience in COVID-19 Patient Care (*n* = 153)
		*n* (%)	*n* (%)
Gender	Man	13(5.9)	19(12.4)
	Woman	206(94.1)	134(87.6)
Age (years)	<30	88(40.2)	82(53.6)
	≥30	131(59.8)	71(46.4)
Marital status	Married	111(50.7)	50(32.7)
	Single and other	108(49.3)	103(67.3)
Education	Associate’s degree	47(21.5)	31(20.3)
	Bachelor’s degree	157(71.7)	112(73.2)
	Graduate school or higher	15(6.8)	10(6.5)
Total clinical period (month)	<36	41(18.7)	48(31.4)
	36–60	30(13.7)	26(17.0)
	>60	148(67.6)	79(51.6)
The current department period (month)	<36	117(53.4)	85(55.6)
	≥36	102(46.4)	68(44.4)
Work department	General ward	159(72.6)	99(64.7)
	Special ward	60(27.4)	54(35.3)
Number of infection control education	1	60(27.4)	38(24.8)
	2	48(21.9)	33(21.6)
	3 or more times	111(50.7)	82(53.6)
Education hours (min per session)	Less than 30	89(40.6)	55(35.9)
	30–60	22(10.0)	23(15.0)
	More than 60	108(49.3)	75(49.0)
Main learning methods	Lecture	44(20.1)	34(22.2)
	Online education	23(10.5)	19(12.4)
	Clinical training	131(59.8)	72(47.1)
	Other	21(9.6)	28(18.3)
Effective learning methods	Lecture	28(12.8)	29(19.0)
	Audiovisual education	46(21.0)	33(21.6)
	Lecture and training	131(59.8)	78(51.0)
	Online education and others	14(6.4)	13(8.5)

**Table 2 healthcare-09-01043-t002:** Analysis of educational needs and priority of education among participants with prior experience in COVID-19 patient care.

Contents	Importance	Confidence	Difference	t	*p*-Value	Borich Priority	Rank	The Locus for Focus Model
	M ± SD	M ± SD	M ± SD					
Initial response to communicable respiratory infection	4.75 ± 0.51	3.96 ± 0.80	0.79 ± 0.86	13.46	<0.001	3.75	1	1st quadrant (HH)
Management of aerosol-generating procedures in patients with communicable respiratory infection	4.61 ± 0.57	3.83 ± 0.90	0.77 ± 0.91	12.57	<0.001	3.58	2	2nd quadrant (LH)
Reporting of patients with communicable respiratory infection and death of patient	4.5 ± 0.68	3.82 ± 0.93	0.67 ± 0.98	10.19	<0.001	3.04	3	2nd quadrant (LH)
Characteristics of communicable respiratory infections	4.80 ± 3.45	3.97 ± 0.77	0.82 ± 0.49	3.50	0.001	2.82	4	1st quadrant (HH)
Comunicable respiratory infection patients, nurse and medical staff movements (including admission to other hospitals and intensive care units, admission to isolation rooms, and check-out)	4.64 ± 0.56	4.04 ± 0.80	0.59 ± 0.86	10.24	<0.001	2.78	5	3rd quadrant (LL)
Guidelines for attention to patients with communicable respiratory infection	4.71 ± 0.49	4.13 ± 0.75	0.57 ± 0.72	11.80	<0.001	2.71	6	4th quadrant (HL)
Criteria for the confirmation and releasing patients from isolation with communicable respiratory infection	4.66 ± 0.58	4.08 ± 0.83	0.57 ± 0.83	10.21	<0.001	2.68	7	4th quadrant (HL)
Methods for assigning and operating isolation rooms for patients with communicable respiratory infection (including hospitalization rules)	4.61 ± 0.61	4.05 ± 0.80	0.55 ± 0.87	9.44	<0.001	2.57	8	3rd quadrant (LL)
Testing and management of patients with communicable respiratory infection	4.64 ± 0.55	4.12 ± 0.79	0.53 ± 0.78	9.97	<0.001	2.46	9	3rd quadrant (LL)
Recommended range and application of personal protective equipment	4.74 ± 0.49	4.24 ± 0.72	0.49 ± 0.71	10.33	<0.001	2.36	10	4th quadrant (HL)
Disinfection policy for equipment and sickroom environment (including disinfection of sickroom after discharge, laundry management, and medical waste management)	4.63 ± 0.62	4.16 ± 0.73	0.47 ± 0.82	8.65	<0.001	2.22	11	3rd quadrant (LL)
Management of family members, visitors, and caregivers when patients with communicable respiratory infection are hospitalized	4.59 ± 0.59	4.17 ± 0.78	0.42 ± 0.77	8.09	<0.001	1.95	12	3rd quadrant (LL)

M = Mean, SD = Standard deviation, HH = High discrepancy and high importancy, HL = High discrepancy and low importancy, LH = Low discrepancy and high importancy, LL = Low discrepancy and low importancy.

**Table 3 healthcare-09-01043-t003:** Analysis of educational needs and priority of education among participants without prior experience in COVID-19 patient care.

Contents	Importance	Confidence	Difference	t	*p*-Value	Borich Priority	Rank	The Locus for Focus Model
	M ± SD	M ± SD	M ± SD					
Management of aerosol-generating procedures in patients with communicable respiratory infection	4.52 ± 0.66	3.39 ± 0.94	1.12 ± 1.05	13.69	<0.001	5.04	1	2nd quadrant (LH)
Initial response to communicable respiratory infection	4.70 ± 0.52	3.74 ± 0.79	0.96 ± 0.88	13.50	<0.001	4.49	2	2nd quadrant (LH)
Reporting of patients with communicable respiratory infection and death of patient	4.55 ± 0.64	3.59 ± 0.92	0.96 ± 0.92	11.59	<0.001	4.34	3	2nd quadrant (LH)
Comunicable respiratory infection patients, nurse and medical staff movements (including admission to other hospitals and intensive care units, admission to isolation rooms, and check-out)	4.61 ± 0.52	3.74 ± 0.87	0.87 ± 0.91	11.86	<0.001	4.02	4	2nd quadrant (LH)
Testing and management of patients with communicable respiratory infection	4.67 ± 0.52	3.80 ± 0.85	0.86 ± 0.93	11.44	<0.001	4.00	5	1st quadrant (HH)
Disinfection policy for equipment and sickroom environment (including disinfection of sickroom after discharge, laundry management, and medical waste management)	4.66 ± 0.57	3.82 ± 0.83	0.83 ± 0.96	10.74	<0.001	3.87	6	4th quadrant (HL)
Methods for assigning and operating isolation rooms for patients with communicable respiratory infection (including hospitalization rules)	4.60 ± 0.56	3.77 ± 0.87	0.83 ± 0.93	10.95	<0.001	3.79	7	3rd quadrant (LL)
Criteria for the confirmation and releasing patients from isolation with communicable respiratory infection	4.63 ± 0.55	3.82 ± 0.82	0.81 ± 0.91	11.06	<0.001	3.76	8	4th quadrant (HL)
Recommended range and application of personal protective equipment	4.74 ± 0.48	3.95 ± 0.84	0.79 ± 0.87	11.13	<0.001	3.72	9	4th quadrant (HL)
Management of family members, visitors, and caregivers when patients with communicable respiratory infection are hospitalized	4.69 ± 0.51	3.92 ± 0.80	0.76 ± 0.89	10.57	<0.001	3.56	10	4th quadrant (HL)
Guidelines for attention to patients with communicable respiratory infection	4.68 ± 0.52	3.92 ± 0.77	0.75 ± 0.81	11.55	<0.001	3.52	11	4th quadrant (HL)
Characteristics of communicable respiratory infections	4.52 ± 0.55	3.79 ± 0.77	0.73 ± 0.72	12.48	<0.001	3.29	12	4th quadrant (HL)

M = Mean, SD = Standard deviation, HH = High discrepancy and high importancy, HL = High discrepancy and low importancy, LH = Low discrepancy and high importancy, LL = Low discrepancy and low importancy.

## Data Availability

The data presented in this study are available on request from the corresponding author.

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
