# Peer review of "Educational Needs of Nurses for Respiratory Communicable Infectious Disease Care: A Cross-Sectional Descriptive Study"

_healthcare, 2021, doi:10.3390/healthcare9081043_

Round 1

Reviewer 1 Report

Both in the abstract and in the discussion, the sentence regarding the purpose (aim)of the work should be shifted. At the end of the abstract. And from the discussion it should be moved to end of introduction.

Abstract should summarize the article. Currently, it is not suited to it.

In the introduction, a few opening lines are unnecessary because they do not concern the topic of the work. The introduction should focus more on the intended topic of work. There is no support in the international discussion here. it should be reinforced.

The method is not transparent. I suggest dividing it into the following sections, starting with study location, population / participant, instrument, collection, annaylys. What about Ethical considerations? There is no statment about this

results: 3.1 part of this should go to method
Why table 1 is after 3.2 when it should go before?

Discusion: this section needs a lot of improvments. There are no references to better research and information to compare it to.
Read the examples below and use them to strengthen this section
https://doi.org/10.1016/j.pedn.2021.01.027
https://doi.org/10.1016/j.ijdrr.2021.102195
https://doi.org/10.1371/journal.pone.0244488
https://doi.org/10.1186/s12904-021-00738-x

Limitations should be separeted from conclusion and conclusion should be expanded. Also do you have some recomendations for healthcatre providers or policymaking?

Both in the abstract and in the discussion, the sentence regarding the purpose of the work should be shifted. At the end of the abstract. And in the discussion it should be moved to introduction.

Abstract should summarize the article. Currently, it is not suited to it.

In the introduction, a few opening lines are unnecessary because they do not concern the topic of the work. The introduction should focus more on the intended topic of work. There is no support in the international discussion here. it should be reinforced.

The method is not transparent. I suggest dividing it into the following sections, starting with study location, population / participant, instrument, collection, annaylys. What about Ethical considerations? There is no statment about this

results: 3.1 part of this should go to method
Why table 1 is after 3.2 when it should go before?

Discusion: this section needs a lot of improvments. There are no references to better research and information to compare it to.
Read the examples below and use them to strengthen this section
https://doi.org/10.1016/j.pedn.2021.01.027
https://doi.org/10.1016/j.ijdrr.2021.102195
https://doi.org/10.1371/journal.pone.0244488
https://doi.org/10.1186/s12904-021-00738-x

Limitations should be separeted from conclusion and conclusion should be expanded. Also do you have some recomendations for healthcatre providers or policymaking?

23 references are definitely not enough when it comes to the power of scientific work. They are especially missing in the discussion

Author Response

Comments from the editor and reviewers, and author’s response

Dear Reviewers and Editor:

Thank you for reviewing the paper “Educational needs of nurses for respiratory com-municable infectious disease care: A cross-sectional descriptive study” submitted for review for publication in the Healthcare. I would also like to thank the anonymous reviewers for their comprehensive and helpful comments. The manuscript has been revised accordingly. We look forward to hearing from you. Please let us know if you have any further questions. Thank you for your kind attention.

Sincerely,

Authors

[Reviewer 1: Comments and Suggestions for Authors]

Comment 1: Both in the abstract and in the discussion, the sentence regarding the purpose (aim) of the work should be shifted. At the end of the abstract. And from the discussion it should be moved to end of introduction.

Response 1: We have revised the contents according to your comments.

Comment 2: Abstract should summarize the article. Currently, it is not suited to it.

Response 2: We have revised the contents of abstract according to your comments.

Comment 3: In the introduction, a few opening lines are unnecessary because they do not concern the topic of the work. The introduction should focus more on the intended topic of work. There is no support in the international discussion here. it should be reinforced.

Response 3: We have revised the contents according to your comments.

Comment 4: The method is not transparent. I suggest dividing it into the following sections, starting with study location, population / participant, instrument, collection, annaylys. What about Ethical considerations? There is no statment about this

Response 4: We have revised the title in methods section according to your comments. However, we have already described Ethical considerations in page 5 line 156-157. (this study was approved by the institutional review board (KHSIRB-20-330(EA)

Comment 5: results: 3.1 part of this should go to method
Why table 1 is after 3.2 when it should go before?

Response 5: We have revised the contents according to your comments.

Comment 6: Discusion: this section needs a lot of improvments. There are no references to better research and information to compare it to.
Read the examples below and use them to strengthen this section
https://doi.org/10.1016/j.pedn.2021.01.027
https://doi.org/10.1016/j.ijdrr.2021.102195
https://doi.org/10.1371/journal.pone.0244488
https://doi.org/10.1186/s12904-021-00738-x

Response 6: We have added references and revised the contents according to your comments.

Comment 7: Limitations should be separeted from conclusion and conclusion should be expanded. Also do you have some recomendations for healthcatre providers or policymaking?

 Response 7: We have revised the contents according to your comments.

Reviewer 2 Report

1) I find the study interesting and current. The analysis of the nurses skills who manage COVID 9 patients, is really crucial! I therefore believe that the proposed topic and the analyses carried out by the authors report very useful results, both for the patient  safety both for nursing students and the developments on the curriculum  (as the authors underline in their conclusions).

2) I find that the background is well written. It is well written about the research problem and was reported a previous study in which the skills of Korean nurses were analyzed.

3) the materials and methods are well described. Clear! The only thing is that I would report the study design. As I said in the small report sent to your journal.

4) the results are well defined and relevant as I said in point 1.

5) the discussion is well articulated and well argued.

6) the only thing is that in the conclusions section, I would report in the limits section, that the results refer to self-report data with all the limits that this entails.

Round 2

Reviewer 1 Report

Thanks to authors for accept my suggestions